# Humor Coping Reduces the Positive Relationship between Avoidance Coping Strategies and Perceived Stress: A Moderation Analysis

**DOI:** 10.3390/bs13020179

**Published:** 2023-02-16

**Authors:** Luca Simione, Camilla Gnagnarella

**Affiliations:** 1Istituto di Scienze e Tecnologie della Cognizione, ISTC, CNR, 00185 Rome, Italy; 2ASST Valtellina e Alto Lario, 23100 Sondrio, Italy

**Keywords:** humor, coping, approach, avoidance, Brief-COPE, moderation, stress, COVID-19, pandemic

## Abstract

Humor is considered an adaptive coping strategy as it could reduce the burden of perceived stress and increase positive emotional states when dealing with stressful situations. Humor has been reported in several models as a rather independent strategy that can be correlated with both approach-based coping strategies and avoidance-based coping strategies. Humor can be defined as a hedonistic escapism strategy that would work better in the presence of unpredictable or uncontrollable stressors, such as the spread of the COVID-19 pandemic and its related confinement measures. Therefore, during such a stressful event, humor would have increased the positive effect of the approach coping style on mental health and reduced the negative effect of the avoidance coping style. Based on this hypothesis, we conducted a cross-sectional study with a moderation analysis in which we assessed the interaction of humor with both approach-based and avoidance-based coping styles on perceived stress in a large sample of Italian participants collected in April and May 2021. Despite some limitations related to sampling and study design, the results obtained partially support our hypothesis, as we observed that humor had a significant moderating effect on the relationship between avoidance coping and psychological distress, with a reduction of perceived stress while using such a coping style in the presence of a medium to high level of humor. On the other hand, we did not observe a significant moderating effect of humor on the relationship between the approach coping style and perceived stress. In general, our results support the beneficial effect of humor on mental health and highlight a special role for humor as a moderator of other coping strategies.

## 1. Introduction

Humor can be defined as the ability to perceive or express the humorous aspects of a situation [1]. Therefore, when dealing with stressful events or situations, humor can act as a useful coping strategy to reduce stress and increase positive affect. Humor is generally considered an adaptive coping strategy [2], that is, a strategy that has been proven to be effective in reducing perceived stress in the presence of stressful events. Humor is also effective in increasing psychological well-being and reducing psychological symptoms. It is exploited as a therapeutic strategy in many interventions [3], in particular in the field of positive psychology [4]. Mechanisms by which humor works have been investigated in many reviews and meta-analyses [5,6,7,8]. In general, the beneficial effect of humor is considered in light of its ability to relieve stress, anxiety, depression, and psychological distress, while increasing optimism, social relationships, and life satisfaction. Humor also increases positive affect states and helps to maintain them over time.

Humor is usually positively correlated with both approach coping strategies and avoidant coping strategies [9,10]. The first group of strategies, collectively grouped as the approach coping style, is related to a reduced stress and increased mental health in the long run. On the other hand, the second group of strategies, clustered as the avoidant coping style, increases the burden of stress and could lead to poor mental health [2]. Approach coping includes strategies such as active coping, positive reframing, and planning, while avoidance coping includes denial, behavioral disengagement, and distraction. While humor has been reported in association with all these strategies [11,12,13,14], it is largely considered an independent coping strategy, which does not cluster with the avoidance and approach coping styles [15]. Therefore, humor seems to play a special role in determining the coping resource of an individual. Moreover, humor style is not a monolithic concept, but different forms of humor can be individuated and more related to active coping or hedonism/avoidance [16]. Thus, humor can be applied in different ways to cope with stress, and in conjunction with other coping strategies, could lead to different results. 

In a recent review article, Stanisławski proposed the construct of hedonistic disengagement, defined as “a combination of problem avoidance and positive emotional coping. Hedonic disengagement involves the avoidance of information on the problem and a strong tendency to maintain momentary well-being” [17]. In this sense, humor would help maintain a good emotional state while avoiding solving the problem at hand; and this strategy has been proposed to be particularly effective with low controllable or uncontrollable stressors, such as the COVID-19 pandemic scenario [18]. During the COVID-19 pandemic, humor coping has been associated with approach-based coping strategies, such as positive reframing and acceptance [14,19]. Furthermore, there is evidence that humor as a coping strategy was correlated with reduced stress related to COVID-19, especially in healthcare workers [10], who were at higher risk of increased psychological distress compared to the general population due to proximity to infected people [20,21]. In such a scenario, a high level of preoccupation with the stressor could lead to more negative outcomes [22]. Therefore, to cope with stress due to the COVID-19 pandemic, a highly uncontrollable situation [23], humor could buffer the negative effect of avoiding coping strategies while enhancing the effect of approaching and active coping strategies, the latter of which could eventually lead to increased perceived stress, due to the low controllable nature of this stressor.

To test this hypothesis, we conducted a moderation study in which we analyzed the strategies for coping with pandemic stress used by a large data set of Italian respondents, as well as their relationship with perceived stress. In general, we expected that humor coping would be correlated with reduced perceived stress. However, our main hypothesis was that humor coping would moderate the relationship between perceived stress with both approach and avoidant coping styles. In the first case, humor would contrast the negative effect of dealing with an uncontrollable stressor when using approach-oriented strategies. In the second case, it would contain the eventual negative effect of the avoidance tendency, usually related to negative mental health outcomes, by giving a stable positive affective state over time. 

## 2. Materials and Methods

### 2.1. Participants

The sample for this study was collected in April and May 2020 from the general Italian population. This sample was collected for previous independent projects on the impact of COVID-19 on mental health that we conducted during that period and then merged into a large sample. The inclusion criteria were to be Italians, in the range of 18–65, with an internet connection. The only exclusion criterion was being positive with COVID-19 at the time of compilation. All participants were volunteers and were recruited using the random sampling method. The final sample for the analysis included 1413 participants. 1288 participants were females and 125 were males, the mean age was 39.97 years (SD = 7.95), and the mean education level in years was 14.34 (SD = 3.63). We obtained compiled informed consent from all participants before completing the questionnaires. All data was collected in a completely anonymous format. Ethical approval for this study was granted by the Research Ethics and Integrity Committee of CNR, and all procedures performed were in accordance with the ethical standards of the Declaration of Helsinki of 1964.

### 2.2. Materials and Procedure

All participants completed a series of online forms, each of which contained a different set of questions or a questionnaire. Among other measures, not presented here as they were outside the scope of this article, participants responded to a set of questions assessing demographic variables, coping strategies applied during the COVID-19 pandemic, and their perceived stress. Demographic variables included some basic information, such as sex, age, and level of education. It also included some information about family composition and working condition, not presented here as it was not consistently assessed for all participants. 

Coping strategies were assessed through the Italian version of Brief-COPE, BC [9,24]. The questionnaire was presented in its situational form, since participants were asked to report how they coped with a specific stressor, i.e., pandemic-related stress, and the items were formatted in the present perfect tense [9]. The BC assesses 14 coping strategies with two items each, for a total of 28 items. Each item reports a coping strategy and participants should indicate how much they rely on that coping strategy on a four-point Likert scale ranging from 1 (“I’ve not been doing this at all”) to 4 (“I’ve been doing this a lot”). The coping strategies evaluated by BC are the following: self-distraction, active coping, denial, substance use, use of emotional support, use of instrumental support, behavioral disengagement, venting, positive reframing, planning, humor, acceptance, religion, and self-blame. For this study, we grouped strategies into two second-order factors or coping styles [10], that is, approach (including active coping, planning, positive reframing, acceptance, use of emotional support, and use of instrumental support) and avoidance (including self-distraction, denial, substance use, behavioral disengagement, venting, and self-blame). In our sample, the reliability analysis obtained satisfactory results for avoidance; Cronbach’s alpha = 0.63 and McDonald’s omega = 0.65, and approach, Cronbach’s alpha = 0.81 and McDonald’s omega = 0.81.

Humor coping is usually considered independent of both approach and avoidant styles [25], also with regard to COVID-19 pandemic-related stress [19]. It was assessed with the BC Humor coping scale, which includes two distinct aspects of humor: one linked to problem-focused strategies and active coping (e.g., ‘I make jokes about it”), while the other is linked to a combination of hedonism and avoidance (e.g., “I make fun of the situation”). Therefore, evaluating humor coping with this instrument has the advantage of measuring more coping strategies and styles with a single questionnaire. Moreover, it allowed us to assess both the contribution of the humor coping style per se to well-being and its relationships with the other coping strategies.

Psychological distress was assessed with the 10-item version of the Perceived stress scale, PSS [26,27]. This inventory assesses the presence of perceived stress in the last month and each item describes an overloading or uncontrollable situation or feeling (“In the last month, how often have you felt nervous and stressed?” or “In the last month, how often have you been angered because of things that happened that were outside of your control?”) that should be evaluated on a five-point Likert scale, ranging from 0 (“never”) to 4 (“very often”). PSS showed good reliability in our sample, with Cronbach’s alpha = 0.81 and McDonald’s omega = 0.82. 

### 2.3. Data Analysis

To investigate the relationship between humor, coping styles, and perceived stress, we assessed the relationship between such variables. First, we performed a bivariate Pearson’s correlation analysis to verify the general relationship pattern between the variables considered. The effect sizes were interpreted as small, medium, and large, respectively, for values of 0.10, 0.20, and 0.30 [28]. This analysis also included demographic factors of age and level of education, while the difference in psychological variables by sex was assessed using two-sample *t*-tests.

Following this first analysis, we performed the main analysis of this study to assess the interaction between humor and other coping styles on perceived stress. This hypothesis was investigated through a hierarchical multiple linear regression analysis, conducted with a two-step model. The dependent variable was the PSS score, while the predictors were humor, approach, and avoidance coping styles. The first step included all direct effects of the predictors, while in the second step, the humor x approach and the humor x avoidance interaction terms were also included. The analysis was controlled for covariates of sex, age, and level of education. For each model, the unstandardized regression coefficients (*b*) and the standardized (*β*) regression coefficients were reported along with their 95% confidence intervals, as well as the statistical test for significance. Furthermore, the general fit of the model was reported as *R*^2^ and adjusted *R*^2^, and statistical tests were performed to assess the deviation of *R^2^* from zero and the increase in *R^2^* from step 1 to step 2. 

A significant interaction term would be further investigated through a simple slope analysis. The relationship between the predictor (approach or avoidant coping) and the dependent variable (perceived stress) was evaluated at three levels of the moderator (humor), that is, when the moderator was low (−1 SD from the mean), average (mean), or high (+1 SD from the mean). For each slope test, the coefficient of this relationship was reported along with its 95% confidence intervals and the level of significance. Before running the moderation analysis, both the predictors and the moderator were mean centered to reduce estimation and multicollinearity problems [29] and make the regression coefficients more meaningful [30].

The second-step model included a total of three predictors, three covariates, and two interaction terms. Therefore, considering the sample size of 1413 participants, we obtained an event per variable of approximately 177, when the minimum suggested level is about 20 [31]. Furthermore, a power analysis conducted with GPower 3.1 [32] indicated that, with a sample size of 1400, we could detect even a very small effect with a power of 0.98 for both the single regression coefficient and the overall regression model statistic (deviation of *R*^2^ from zero and increase in *R*^2^).

All analyses were performed with Jamovi [33], and R statistical software [34]. 

## 3. Results

### 3.1. Descriptive and Preliminary Analysis

At a descriptive level, the use of humor coping during the pandemic was moderately reported in our sample. The mean score on the humor scale was 3.586 (*SD* = 1.342), and the median value was 3. Considering that the scale score was computed as the sum of two items evaluated on a Likert scale ranging from 1 to 4, the expected median point should have been 5 on a score ranging from 2 to 8. The use of approach coping was on average 30.778 (*SD* = 6.099) and the use of avoidance coping was 23.038 (*SD* = 4.484), with both scores ranging from 12 to 48. Therefore, participants reported a greater use of approach coping strategies than avoidance coping strategies. The PSS score was on average 20.238 (*SD* = 6.629), which is considered indicative of mild stress. For comparison, the mean value reported in the validation study for the Italian version [26] was lower, as it was 15.2 and 16.3, respectively, for males and females. 

We then evaluated the relationship between demographic variables and psychological factors. Next, we performed Pearson’s bivariate correlation analysis for age and education level, and a series of independent sample Welch’s t-tests for sex. The correlation analysis (see Table 1) on age reached significance for its relationship with PSS, *r* = −0.138, *p* < 0.001, BC Humor, *r* = 0.058, *p* < 0.05, and avoidance, *r* = −0.089, *p* < 0.001. The correlation analysis on the level of education reached significance for its relationship with PSS, *r* = −0.088, *p* < 0.001, BC Humor, *r* = 0.061, *p* < 0.05, approach, *r* = 0.133, *p* < 0.001, and avoidance, *r* = −0.064, *p* < 0.05. Regardless of their significance, all these correlation coefficients had a small to negligible effect size. The analysis of sex revealed that, compared to males, females reported a higher level of perceived stress, M_females_ = 20.50, M_males_ = 17.54, *t*(147) = −4.59, *p* < 0.001, *d* = −0.440, lower level of BC Humor, M_females_ = 3.98, M_males_ = 3.55, *t*(142) = 2.96, *p* < 0.01, *d* = 0.296, higher level of approach, M_females_ = 31.25, M_males_ = 29.56, *t*(150) = −3.00, *p* < 0.01, *d* = −0.279, and a higher level of avoidance, M_females_ = 23.23, M_males_ = 21.06, *t*(146) = −5.00, *p* < 0.001, *d* = −0.479. Overall, this analysis showed that women reported more stress during the COVID-19 pandemic and tended to apply more coping strategies, both approaching and avoiding, but less humor. As we obtained small but significant results for all demographic variables, they were included as covariates in the subsequent regression analysis to control for their influence.

### 3.2. Correlation Analysis

The pattern of correlation between the psychological variables was consistent with expectations. The perceived stress assessed with PSS was negatively correlated with BC Humor, *r* = −0.088, *p* < 0.001, and approach coping, *r* = −0.100, *p* < 0.001; and positively with avoidant coping, *r* = 0.399, *p* < 0.001. BC Humor was positively correlated with both approach, *r* = 0.247, *p* < 0.001, and avoidance coping, *r* = 0.138, *p* < 0.001; and approach and avoidance coping styles were also correlated, *r* = 0.307, *p* < 0.001. In general, this analysis showed that participants reported using a combination of different coping styles, and that humor coping was positively correlated with both approach and avoidance styles. Furthermore, humor appears to have only a small negative correlation with perceived stress, suggesting that a more complex relationship model could better explain its expected positive relationship with well-being.

### 3.3. Hierarchical Regression Analysis

The main hypothesis of this study was tested with a hierarchical regression model. The model coefficients are reported in Table 2. The first step of the model included all direct effects of BC Humor, approach, and avoidance on PSS while controlling for the covariates of sex, age, and educational level. Overall, the fit of the model was significant, *R*^2^ = 0.235, adjusted *R*^2^ = 0.232, *F*(6,1406) = 72.1, *p* < 0.001. As expected, the two main coping styles had an opposite relationship with PSS, as the approach was linked to a decrease in PSS while avoidance to an increase in PSS. Humor also had a significant direct effect on PSS, as it was related to a lower level of perceived stress.

Step 2 of the model included the interactions of humor with approach and avoidance. The fit of the model increased significantly from Step 1, with Δ*R*^2^ = 0.005, *p* < 0.01. This increase in model fit was qualified by the significant humor x avoidance interaction, while the humor x approach interaction did not reach significance. 

The significant interaction was further analyzed using simple slope analysis. The analysis tested the relationship between avoidance and PSS when the humor score was low, average, or high. This relationship was significant at all three levels of the moderator humor but decreased in size. The result of this analysis is reported in Figure 1. When humor was low, the relationship between avoidance and PSS was positive, *b* = 0.715, *SE* = 0.051, *p* < 0.001. When the humor level was on average, this relationship was again positive, but decreased in magnitude, *b* = 0.623, *SE* = 0.036, *p* < 0.001; and it further decreased when the humor was high, *b* = 0.531, *SE* = 0.048, *p* < 0.001. This pattern of results supports the hypothesis that the presence of humor could be linked to a reduction in the negative effect of maladaptive coping strategies on psychological well-being via increased perceived stress, but only partially.

## 4. Discussion

In this paper, we report the results of a moderation analysis in support of the hypothesis that humor coping interacts with other coping strategies, such as the approach-based and avoidance-based strategies assessed with the Brief COPE inventory. We hypothesized that using the approach coping style would be related to an increased reduction in perceived stress when humor was present, and that using the avoidance coping style would be related to a decreased level of stress when humor was also present. We also hypothesized, in general, that humor would be an adaptive strategy to cope with the uncontrollable stress caused by the COVID-19 pandemic and its related confinement measures, such as lockdowns and quarantines. Therefore, we hypothesized that the use of humor coping would be negatively correlated with psychological stress.

In general, our results partially confirmed these hypotheses. In fact, we found a negative but weak correlation between humor and perceived stress. In addition, in the first step of the regression analysis, increased use of humor coping predicted a lower perceived stress score. However, this effect had a standardized coefficient lower than both the negative coefficient of the approach coping style and the positive coefficient of the avoidant coping style. Thus, humor appeared as an adaptive coping strategy in the pandemic scenario, but its suggested positive effect on stress appears to be limited compared to other coping strategies. This poor correlation between humor coping and stress could be explained, as maladaptive forms of humor could be exploited as a coping strategy [35]. As we evaluated humor using the two-item scale of the Brief COPE, which is very limited with respect to more refined inventories such as the Coping Humor scale [36], this hypothesis could not be further investigated in our data set.

Despite the small effect size, the negative correlation obtained between humor and psychological distress has already been reported during the COVID-19 pandemic [10,19], even if there are some inconsistent results in the literature [37]. In a sense, humor could be referred to as an escape strategy, in which individuals avoid directly managing the problem that causes stress while trying to maintain a positive attitude. This resembles the notion of hedonistic escapism introduced by Stanisławski [17], which includes some forms of coping based on humor. Thus, in the unpredictable and uncontrollable pandemic scenario [23], an avoidant but positive strategy, such as humor coping, could be more effective than other coping strategies that cannot be properly exploited, such as seeking emotional support or seeking help from others [14]. Consistently, Tahara et al. [38] reported how healthcare workers in Japan relied mainly on avoidance or escape strategies to manage pandemic-related stress and made behavioral efforts to avoid or escape problems. This also seems to suggest that humor would be more effective for those people who were more exposed to COVID-19, such as healthcare workers [10,38]. A recent quasi-experimental study on healthcare workers [39] reported that attending humor-based training decreased gelotophobia, which in turn has been linked to decreased life satisfaction and increased stress. Thus, humor could be a valid aid in stressful contexts for increasing psychological well-being and coping attitudes. 

The moderation analysis also partially confirmed our hypothesis. We obtained a significant humor x avoidance coping interaction, while the moderator effect of humor on the relationship between approach coping and perceived stress did not reach significance. This latter finding suggests that using humor coping could have an additive, not a multiplicative, effect on mental health while using an already adaptive coping style, such as approaching coping. Humor coping has often been reported in combination or strongly correlated with many other coping strategies that are generally considered adaptive, such as problem-solving, acceptance, and positive reframing [13,14,40,41]. Humor coping itself could work through positive reinterpretation and reframing of the stressor [17], partially overlapping the working mechanisms of the approach-based coping strategies. A possible explanation for this lack of interaction effect could be found in the outcome variable. We inferred perceived stress to be an indicator of psychological distress, but did not include a measure of psychological well-being in terms of quality of life or subjective satisfaction/happiness. Therefore, it is possible that humor would better combine with approach coping in predicting psychological well-being; as also suggested by previous data showing how it influenced quality of life during the COVID-19 pandemic, but not anxiety or depression [37]. Again, humor coping could work better for developing and maintaining a positive outlook on situations [42,43,44], rather than for directly buffering negative emotions.

This point also supports the main finding of this article, that is, regarding the interaction between humor and avoidance to reduce the negative impact of the latter on mental health. We found that the avoidance coping style was correlated with an increase in perceived stress during the COVID-19 pandemic, but that the entity of such correlation was reduced in the presence of medium to high levels of humor. As the COVID-19 scenario was characterized by uncontrollability and unpredictability, a hedonistic escapism strategy would have been effective in reducing perceived stress while using other avoidance-based strategies [17,45]. However, this strategy seems to only be capable of reducing the maladaptive effect of avoidance-based coping strategies, but not capable of eliminating it or transforming such strategies into adaptive ones. Thus, caution should be applied in considering the practical implications of this result. It suggests that, in people who rely on avoidance coping to deal with stressors, increasing sense of a humor and the use of a humor coping strategy could alleviate perceived distress and increase positive emotional states, while reducing the negative effects of life stressors. However, this should only be considered as a first step before increasing more adaptive and compelling coping strategies, which could further increase the coping capacity and resilience of individuals facing stressors [3]. In the same vein, increasing sense of humor and diffusing the use of humor to cope with stress in workers could buffer the effects of workplace stressors, especially in complex and unpredictable scenarios, such as those caused by the worldwide spread of the COVID-19 pandemic [10,45].

Regarding the demographic factors included, we found that females report lower levels of humor coping than males, which is consistent with previous research on the topic [46]. This difference in the use of humor coping seems to be unrelated to the effect of humor coping on psychological well-being, i.e., humor would have the same effect for both males and females who used it as a coping strategy [43]. Female sex was also associated with increased perceived stress and the use of other coping strategies, both approach-based and avoidance-based. We can speculate, with respect to this result, that the relatively less frequent use of humor among females could result in increased psychological distress, and that an increased use of coping strategies in general may be less effective when humor is reduced. Further studies are needed to understand the relationship between sex and humor coping, and whether this could, at least partially, explain the lower levels of psychological distress commonly reported by men. Regarding the other demographic variables, age and educational level were both positively related to humor coping. However, due to the relatively small effect of these correlations and the inconsistent results reported in the literature [7], we were unable to interpret such results reliably.

The results of this study should be interpreted in light of certain limitations. First, our sample mainly includes female participants. Consistent with previous literature [46,47], we also found that male participants reported a higher score for humor coping strategies compared to females. Although we controlled for the effect of sex throughout our analyses, future studies should confirm our findings with a more balanced and representative sample. Another limitation is the lack of measures for psychological well-being. As humor coping primarily increases positive emotional states and perceived well-being [42,43], only considering perceived distress as a measure of psychological status could be limiting. Future studies should carefully consider including both a measure of psychological distress and a measure of psychological well-being to correctly assess the mental health status of participants. The final limitation is due to the situational stressor considered, that is, the COVID-19 pandemic and its related containment measures. This stressor could be considered special or exceptional [18,48], and studies should be carried out on more common stressors and situations in life to confirm our results.

## 5. Conclusions

In this study, we report the results of a moderation analysis in which we showed how the use of humor coping, in combination with avoidance-based coping strategies, is linked to a reduction in perceived stress and, therefore, could favor a better psychological condition. However, the less consistent result concerning the negative relationship of humor to psychological stress, as well as the lack of moderation with approaching coping should suggest caution in interpreting an overall positive and protective effect of humor during stressful situations, such as the COVID-19 pandemic [49]. Even in light of this cautionary note, the results of this study could have practical implications, both in organizational and clinical settings, for showing how to reduce the possible negative effects of maladaptive coping strategies on mental health. Future studies, as described above, are needed to elucidate the effective capacity of humor for modulating the efficacy of the other coping strategies in reducing psychological distress and increasing psychological well-being. With its ability to increase the positive affective state of individuals while also helping to share the psychological burden in the face of unpredictable stressors [45], humor seems to play a peculiar role in coping and, as such, deserves a special place in the study of stress and stress management.

## Figures and Tables

**Figure 1 behavsci-13-00179-f001:**
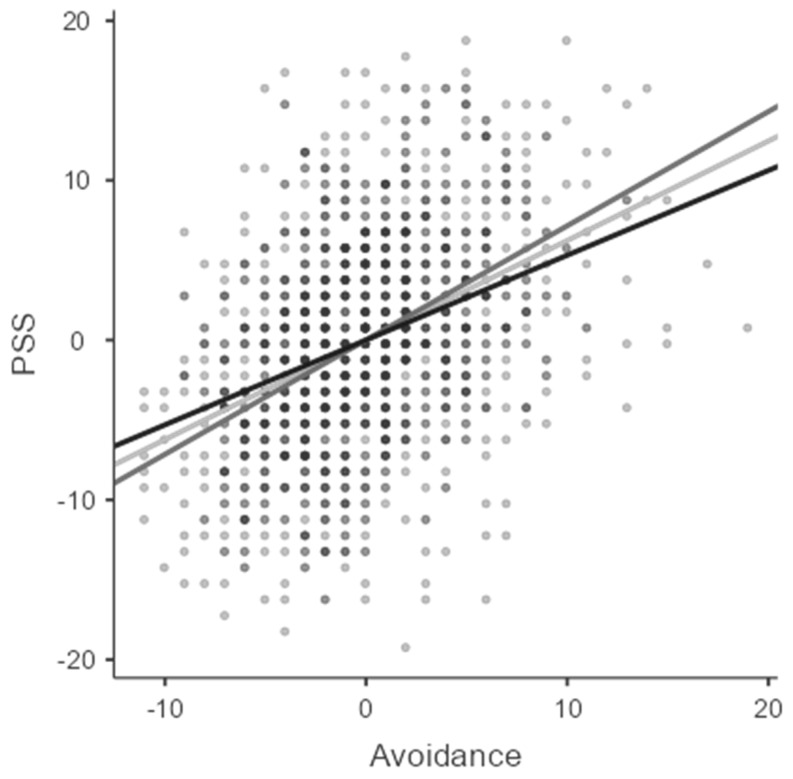
Simple slope analysis for the moderation effect of BC Humor on the relationship between avoidance and PSS. The three lines represent the regression line of the avoidance-PSS relationship when the humor score was average (light gray), low (dark gray), or high (black).

**Table 1 behavsci-13-00179-t001:** Pearson’s correlation coefficients.

	Age		Education Level		PSS		BC Humor		Approach	
PSS	−0.138	**	−0.088	**	—					
BC Humor	0.058	*	0.061	*	−0.088	**	—			
Approach	−0.011		0.133	**	−0.100	**	0.247	**	—	
Avoidance	−0.089	**	−0.064	*	0.399	**	0.138	**	0.307	**

Note. PSS = perceived stress scale; BC = brief COPE. * *p* < 0.05, ** *p* < 0.001.

**Table 2 behavsci-13-00179-t002:** Hierarchical regression analysis with the PSS score as a dependent variable.

Step 1: Direct Effects	*b*	*SE*	*Lower CI*	*Upper CI*	*t*	*p*	*β*
Intercept	15.555	1.504	12.604	18.506	10.339	<0.001	-
Sex	1.429	0.558	0.334	2.524	2.561	0.011	0.061
Age	−0.072	0.020	−0.110	−0.033	−3.635	<0.001	−0.086
Education level	−0.029	0.043	−0.114	0.056	−0.671	0.502	−0.016
BC Humor	−0.416	0.120	−0.651	−0.180	−3.456	<0.001	−0.084
Approach	−0.244	0.028	−0.299	−0.190	−8.831	<0.001	−0.225
Avoidance	0.684	0.037	0.611	0.756	18.513	<0.001	0.462
**Step 2: Moderation Effects**	** *b* **	** *SE* **	** *Lower CI* **	** *Upper CI* **	** *t* **	** *p* **	** *β* **
BC Humor × Approach	−0.004	0.020	−0.043	0.035	−0.203	0.839	−0.005
BC Humor × Avoidance	−0.073	0.026	−0.124	−0.023	−2.833	0.005	−0.067

Note. BC = brief COPE. *b* indicates the unstandardized coefficients, *β* indicates the standardized coefficients, *SE* indicates the standardized errors, *Lower CI* and *Upper CI* indicate, respectively, the lower and upper 95% confidence intervals of the unstandardized coefficients. The sex is coded as 0 = male, 1 = female. Age and level of education are expressed in years.

## Data Availability

Data for this study will be made available to other researchers upon reasonable request.

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
