# Peer review of "Humor Coping Reduces the Positive Relationship between Avoidance Coping Strategies and Perceived Stress: A Moderation Analysis"

_behavsci, 2023, doi:10.3390/bs13020179_

Round 1
Reviewer 1 Report
The manuscript under review examines the role of humor coping with regard to broder coping strategies in a large sample of > 1,000 participants. The findings met the expectations well and the analyses are sound. My impression is that the findings will contribute to the knowledge to field. The manuscript fits well into the scope of the journal and the SI.
(1) Please check the language for over-interpretations. Considering that your data are of cross-sectional data, causality cannot be studied and statements such as "x buffers y" (see title and throughout manuscript) or "effects of age and education" etc. cannot be made reliably.
(2) l. 35: "a strategy that actually works" sounds colloquial
Same sentence; please revise structure
(3) l. 191: Please revise "measured" to "assessed" since you collected self-reports of health
(4) l. 232: Please clarify the effect size of the power analysis
(5) I appreciate that authors evaluate statistical significance independently from effect size. However, I would argue that some correlations are of negligible size, despite significance. Please clarify what your effect size interpretations are (see e.g., Gignac & Szodorai, 2016 for a recommendation of empirically derived effect size interpretations).
Gignac, G. E., & Szodorai, E. T. (2016). Effect size guidelines for individual differences researchers. Personality and Individual Differences, 102, 74-78.
(6) Please revise "sex" to "gender" and "males" to "men" and "females" to "women" because I assume that you collected data on gender instead of sex.
(7) l. 272: Please consider deleting the sentence "In general, this analysis 272 showed how people tended to combine and use multiple coping strategies and that humor 273 coping was reported along with both approach and avoidance styles" because bivariate analyses do not allow such a conclusion that states combinations of predictors.
(8) Please add the abbreviations of the measures to the table notes.
(9) Please provide a comparison of the PSS means between the three groups you analyzed in the moderation analyses to inform about average differences in the dependent variable between the three groups.
(10) l. 317: Please clarify what is meant by "the other coping strategies."
(11) Recently, Vagnoli et al. examined expressions of coping humor in several groups including professional hospital clowns who are faced with stressful situations in the clinic, and they found that coping humor was highest among them. This provides quasi-experimental evidence that coping humor is an important construct when dealing with stressful situations and supports the authors' argument.
Vagnoli, L., Brauer, K., Addarii, F., Ruch, W., & Marangi, V. (2022). Fear of being laughed at in Italian healthcare workers: Testing associations with humor styles and coping humor. Current Psychology, 1-11. https://doi.org/10.1007/s12144-022-03043-9
Author Response
The manuscript under review examines the role of humor coping with regard to broder coping strategies in a large sample of > 1,000 participants. The findings met the expectations well and the analyses are sound. My impression is that the findings will contribute to the knowledge to field. The manuscript fits well into the scope of the journal and the SI.
Thanks for your appreciation of our contribution.
(1) Please check the language for over-interpretations. Considering that your data are of cross-sectional data, causality cannot be studied and statements such as "x buffers y" (see title and throughout manuscript) or "effects of age and education" etc. cannot be made reliably.
The reviewer is right. We eliminated all the causality statements in our text, starting from the title, and give a more reliable and cautious presentation of our results.
(2) l. 35: "a strategy that actually works" sounds colloquial
Same sentence; please revise structure
We fixed this sentence.
(3) l. 191: Please revise "measured" to "assessed" since you collected self-reports of health
Thanks for pointing this out. We amended this point as suggested throughout the manuscript.
(4) l. 232: Please clarify the effect size of the power analysis
We added this information to the text.
(5) I appreciate that authors evaluate statistical significance independently from effect size. However, I would argue that some correlations are of negligible size, despite significance. Please clarify what your effect size interpretations are (see e.g., Gignac & Szodorai, 2016 for a recommendation of empirically derived effect size interpretations).
Gignac, G. E., & Szodorai, E. T. (2016). Effect size guidelines for individual differences researchers. Personality and Individual Differences, 102, 74-78.
Thanks for the nice reference. We added the effect size interpretation in our data analysis section.
(6) Please revise "sex" to "gender" and "males" to "men" and "females" to "women" because I assume that you collected data on gender instead of sex.
We actually asked participants to report their sex, not their gender. Therefore, we prefer to stay with the terms already implied in the manuscript.
(7) l. 272: Please consider deleting the sentence "In general, this analysis 272 showed how people tended to combine and use multiple coping strategies and that humor 273 coping was reported along with both approach and avoidance styles" because bivariate analyses do not allow such a conclusion that states combinations of predictors.
Thanks for this suggestion. As correlation allows to conclude that participants reported to use more than a single coping style, we changed this sentence in “In general, this analysis showed how participants reported to use different coping style together and that also humor coping was positively correlated with both approach and avoidance style”.
(8) Please add the abbreviations of the measures to the table notes.
Done.
(9) Please provide a comparison of the PSS means between the three groups you analyzed in the moderation analyses to inform about average differences in the dependent variable between the three groups.
This point is not applicable, as the simple slope analysis does not allow for group comparison. In fact, this analysis is based on estimating the relationship of two variables when the value of a third one is fixed. This analysis is intended exactly to avoid the sample splitting, which is considered instead a bad practice, as also indicated in influential books on this matter (see Hayes, 2018).
(10) l. 317: Please clarify what is meant by "the other coping strategies."
We clarified this point in the revised manuscript version.
(11) Recently, Vagnoli et al. examined expressions of coping humor in several groups including professional hospital clowns who are faced with stressful situations in the clinic, and they found that coping humor was highest among them. This provides quasi-experimental evidence that coping humor is an important construct when dealing with stressful situations and supports the authors' argument.
Vagnoli, L., Brauer, K., Addarii, F., Ruch, W., & Marangi, V. (2022). Fear of being laughed at in Italian healthcare workers: Testing associations with humor styles and coping humor. Current Psychology, 1-11. https://doi.org/10.1007/s12144-022-03043-9
Thanks for signaling this interesting study. We added it to the discussion when relevant.
Reviewer 2 Report
A good study filling the gap relating humor- stress management issue.
Discussion can still be improved by more contradicting literature.
Author Response
A good study filling the gap relating humor- stress management issue.
Discussion can still be improved by more contradicting literature.
Thanks for your appreciation and your suggestion. We revised our discussion accordingly, adding text reporting contradicting results and trying to explaining our results in relationship with them. Please note that we added in the revised manuscript discussion four references to relevant literature.
Reviewer 3 Report
This study addresses the effects of humor as a coping strategy on psychological distress and coping styles during the COVID-19 pandemic. Using a cross-sectional design, the study analyzed the use of humor during the pandemic in an Italian sample and assessed interactions between humor, approach coping, and avoidance coping. In addition, the effects on psychological distress were calculated using correlation and regression analyses. Although this is a fundamentally interesting topic, the article needs to be revised to provide clarity.
1. Introduction: The introduction has to be modified with regard to its structure. At the moment, the content seems cluttered and the chapters incoherent. I would recommend to start with the content about humor and its importance for stress management/coping. In this context, the paragraphs of lines 85-122 should also be placed directly at the beginning and then linked to COVID-19 pandemic. I would also recommend simplifying the content of the different coping strategies and possibly explaining them independently of the questionnaire used, as this seems confusing when reading. In my opinion, the description of the questionnaire belongs rather to the section on methodology. Overall, the introduction should also be more concise and focus on the content relevant to the research question and therefore be shortened.
2. Methodology: Even though this is stated in the limitations, I find it critical that the article writes that the influence of humor as a coping strategy on mental health was studied, although only perceived stress was measured in this context. I strongly recommend an adjustment of the research question and the hypotheses here. Also, the results should only be interpreted in the “right” understanding of perceived stress in order not to draw wrong conclusions. From a psychological point of view, stress perception is an important influencing factor for mental health, but it is not a direct indicator for the extent of mental health.
3. Results: In terms of the descriptive results, it would be interesting and, in my opinion, relevant to the research question to compare the present sample with the standard sample in terms of the measurements used (Brief-COPE and Perceived Stress Scale). There could already be differences here due to the pandemic.
4. Discussion: With regard to the discussion, from my point of view a more precise interpretation of the socio-demographic data as moderator variables is still missing (e.g., sex, education level and age).
5. The article needs to be revised again with regard to the English language.
Author Response
This study addresses the effects of humor as a coping strategy on psychological distress and coping styles during the COVID-19 pandemic. Using a cross-sectional design, the study analyzed the use of humor during the pandemic in an Italian sample and assessed interactions between humor, approach coping, and avoidance coping. In addition, the effects on psychological distress were calculated using correlation and regression analyses. Although this is a fundamentally interesting topic, the article needs to be revised to provide clarity.
We thank the reviewer for the useful suggestions and for the appreciation of our work.
- Introduction: The introduction has to be modified with regard to its structure. At the moment, the content seems cluttered and the chapters incoherent. I would recommend to start with the content about humor and its importance for stress management/coping. In this context, the paragraphs of lines 85-122 should also be placed directly at the beginning and then linked to COVID-19 pandemic. I would also recommend simplifying the content of the different coping strategies and possibly explaining them independently of the questionnaire used, as this seems confusing when reading. In my opinion, the description of the questionnaire belongs rather to the section on methodology. Overall, the introduction should also be more concise and focus on the content relevant to the research question and therefore be shortened.
Thanks for this nice suggestion. Following it, we greatly reduced the Introduction and removed all the brief-COPE stuff (moved partially in the Methods section). We also removed some unrelated content, especially about Covid-19 pandemic (which is now introduced in its strictly necessary characteristics).
- Methodology: Even though this is stated in the limitations, I find it critical that the article writes that the influence of humor as a coping strategy on mental health was studied, although only perceived stress was measured in this context. I strongly recommend an adjustment of the research question and the hypotheses here. Also, the results should only be interpreted in the “right” understanding of perceived stress in order not to draw wrong conclusions. From a psychological point of view, stress perception is an important influencing factor for mental health, but it is not a direct indicator for the extent of mental health.
We understand your point here and agree with you that perceived stress is not an indicator of the global mental health status of an individual. As a consequence, we referred to perceived stress more than to mental health in the revised manuscript, starting from the revised title.
- Results: In terms of the descriptive results, it would be interesting and, in my opinion, relevant to the research question to compare the present sample with the standard sample in terms of the measurements used (Brief-COPE and Perceived Stress Scale). There could already be differences here due to the pandemic.
Following this suggestion, we added a reference to the PSS score from the validation study. Unfortunately, for the Brief-COPE such values are not reported in the paper by Monzani et al. (2015), who validated the instrument in Italian.
- Discussion: With regard to the discussion, from my point of view a more precise interpretation of the socio-demographic data as moderator variables is still missing (e.g., sex, education level and age).
This is a nice point. We added a new paragraph discussing in particular the effect of sex on humor coping. It is placed just before the limitations (last paragraph of the discussion).
- The article needs to be revised again with regard to the English language.
We extensively review the article also with two AI-based software developed for scientific writing. Unfortunately, we have no possibility to pay for a professional revision as this project received no funding.
Round 2
Reviewer 3 Report
Thank you for the revision according to the recommendations. From my point of view, the article can now be published in its present form.